# Comparing the Real-World Use of Isavuconazole to Other Anti-Fungal Therapy for Invasive Fungal Infections in Patients with and without Underlying Disparities: A Multi-Center Retrospective Study

**DOI:** 10.3390/jof9020166

**Published:** 2023-01-27

**Authors:** Marjorie Vieira Batista, Maria Piedad Ussetti, Ying Jiang, Dionysios Neofytos, Anita Cassoli Cortez, Diego Feriani, Jayr Schmidt-Filho, Ivan Leonardo Avelino França-Silva, Issam Raad, Ray Hachem

**Affiliations:** 1Department of Infectious Diseases, AC Camargo Cancer Center, São Paulo 01509-010, SP, Brazil; 2Transplant Department, Puerta de Hierro University Hospital, 28222 Majadahonda, Spain; 3Department of Infectious Diseases, Infection Control, and Employee Health, The University of Texas MD Anderson Cancer Center, Houston, TX 77030, USA; 4Transplant Department, University Hospital of Geneva, 1205 Geneva, Switzerland; 5Department of Hematology and Cell Therapy, AC Camargo Cancer Center, São Paulo 01509-010, SP, Brazil

**Keywords:** IFI, SCT, disparities, triazoles, isavuconazole, voriconazole, amphotericin B

## Abstract

Invasive fungal infections (IFIs) are a major cause of morbidity and mortality among immunocompromised patients with underlying malignancies and prior transplants. FDA approved Isavuconazole as a primary therapy for Invasive Aspergillosis (IA) and Mucormycosis. This study aims to compare the real-world clinical outcomes and safety of isavuconazole to voriconazole and an amphotericin B-based regimen in patients with underlying malignancies and a transplant. In addition, the response to anti-fungal therapy and the outcome were compared among patients with a disparity (elderly, obese patients, patients with renal insufficiency and diabetes mellitus) versus those with no disparity. We performed a multicenter retrospective study, including patients with cancer diagnosed with an invasive fungal infection, and treated primarily with isavuconazole, voriconazole or amphotericin B. Clinical, radiologic findings, response to therapy and therapy related adverse events were evaluated during 12 weeks of follow-up. We included 112 patients aged 14 to 77 years, and most of the IFIs were classified into definite (29) or probable (51). Most cases were invasive aspergillosis (79%), followed by fusariosis (8%). Amphotericin B were used more frequently as primary therapy (38%) than isavuconazole (30%) or voriconazole (31%). Twenty one percent of the patients presented adverse events related to primary therapy, with patients receiving isavuconazole presenting less adverse events when compared to voriconazole and amphotericin (*p* < 0.001; *p* = 0.019). Favorable response to primary therapy during 12 weeks of follow-up were similar when comparing amphotericin B, isavuconazole or voriconazole use. By univariate analysis, the overall cause of mortality at 12 weeks was higher in patients receiving amphotericin B as primary therapy. However, by multivariate analysis, Fusarium infection, invasive pulmonary infection or sinus infection were the only independent risk factors associated with mortality. In the treatment of IFI for patients with underlying malignancy or a transplant, Isavuconazole was associated with the best safety profile compared to voriconazole or amphotericin B-based regimen. Regardless of the type of anti-fungal therapy used, invasive Fusarium infections and invasive pulmonary or sinus infections were the only risk factors associated with poor outcomes. Disparity criteria did not affect the response to anti-fungal therapy and overall outcome, including mortality.

## 1. Introduction

Patients with hematologic malignancies, mainly acute leukemia, and patients undergoing allogenic stem cell transplantation (SCT), as well as lung transplant, are at a higher risk of developing an invasive fungal infection (IFI) [1,2,3]. *Aspergillus* spp. And *Candida* spp. Are the predominant IFI pathogens in this patient population [4]. However, other fungi such as *Mucolares*, *Scedosporium* spp. and *Fusarium* spp. Are emerging in this scenario. These infections are associated with high mortality rates, in addition to high health costs [4,5,6,7].

In the arsenal of active and efficient antifungal drugs against mold disease, we have polyenes, a model of anti-mold therapy, but they have limited use due to toxicity concerns and the requirement of intravenous administration. Echinocandins, on the other hand, have an excellent safety profile, but are not indicated for the primary treatment of invasive mold infection [8]. Among triazole antifungals, posaconazole has been recently shown to be safe and effective in the treatment of invasive aspergillosis [9]. Voriconazole appears in international guidelines as the primary treatment for invasive aspergillosis [10,11]. However, drug interactions with other drugs frequently used in these patients, such as immunosuppressive drugs, potential liver toxicity and pharmacokinetic variability that require monitoring of serum levels during treatment, all limit its use [10,12].

Isavuconazole, a new azole approved by FDA in 2016 with some different characteristics, such as lower inhibition of CIP3A4, has a lower incidence of adverse events when compared to other antifungals of the same class, mainly concerning hepatic and biliary adverse events; unlike voriconazole or Posaconazole, it is not associated with prolonged QT interval [13,14].

Nowadays, the European Society of Clinical Microbiology and Infectious Diseases/European Confederation of Medical Mycology and ECIL-6 guidelines and Infectious Disease Society of America guidelines recommend isavuconazole or voriconazole as the first-line treatment for IA in high-risk patients, including hematologic malignancy patients [10,11].

The choice of the best antifungal therapy should be optimized according to the characteristics of each patient, taking into account the comorbidities, such as age, renal function, obesity, heart condition, diabetes mellitus and stage of the disease. Additionally, it is important to also consider other drugs in use, the characteristics of the drug, such as pharmacodynamics and pharmacokinetics, PK-PD, drug interactions and toxicity [15,16].

There are scarce data about the comparative safety and efficacy of isavuconazole in the real world, mainly in a setting with disparities. This study aimed to compare the real-world clinical outcomes and safety of isavuconazole to voriconazole and amphotericin B-based regimen in high-risk patients for IFI. In addition, the response to anti-fungal therapy and the outcome were compared among patients with disparities.

## 2. Materials and Methods

### 2.1. Study Design and Data Collection

This multicenter retrospective study was developed after the Institutional Review Board approval in the respective study centers. Three cancer centers in Brazil, Spain and Switzerland were part of the study.

#### 2.1.1. Inclusion Criteria

The patient population to be included will consist of critically ill or immunocompromised patients (e.g., cancer, transplant, etc.) with invasive fungal infections (IFI) and treated with isavuconazole (200 mg every 8 h for 6 doses, then 200 mg daily), voriconazole (6 mg/kg IV every 12 h for 1 d, followed by 4 mg/kg IV every 12 h) and liposomal amphotericin B (3–5 mg/kg/day IV);Patients with a proven, probable or possible invasive fungal infection (e.g., patients who have had a CT scan of the chest suggestive of fungal infection). Please see the definitions below;Age range of patients: 12–86 years of age.

#### 2.1.2. Exclusion Criteria

Patients with less than six months of follow-up;Patients with less than 60% of the data completed.

Demographic and clinical data were collected for all study patients, including age, gender, underlying malignancy, history of stem cell transplant (SCT), presence of graft-versus-host disease (GVHD), solid organ transplant, other immunocompromised condition, obesity (BMI), diabetes mellitus, QT interval (before and after treatment if available) and renal insufficiency, neutropenia status at the onset of infection, persistence of neutropenia during therapy, steroids use, immunotherapy during infection including white blood cell (WBC) transfusion, granulocyte-macrophage colony stimulating factor (GM-CSF), granulocyte colony stimulating factor (G-CSF), and interferon-gamma (IFN-ã), intensive care unit (ICU) stay and need for mechanical ventilation. Type and site of either definite or probable IA along with prior antifungal prophylaxis, breakthrough infection and antifungal therapy were also collected. Outcome information, including response to therapy, all-cause mortality, IA-attributable mortality and adverse events, were also collected. All epidemiological and clinical data were collected using secure, standardized forms and stored in an analytical file system (RedCap).

### 2.2. Definitions

An invasive fungal infection (IFI) was defined according to the revised European Organization for Research and Treatment of Cancer/Mycosis Study Group definitions [17], and a proven or definite mold infection was defined as documented histopathologic and microbiological evidence of mold infection in a tissue biopsy or needle aspiration specimen from a normally sterile site (excluding bronchoalveolar lavage fluid, cranial sinus cavity and urine) or recovery of a mold by blood culture in the context of a compatible infectious diseases process. A probable mold infection was defined by the presence of at least one mycological criteria (cytology, culture of sites that are normally not sterile or detection of antigen or cell-wall constituents), along with one host factor (recent absolute neutrophil count [ANC] < 500 cells/mL, allogeneic stem cell transplant, T-cell immune suppressant therapy, or prolonged corticosteroid use and one clinical criterion (nodules, cavitary, or ground glass opacities found on pulmonary computed tomography [CT]; tracheobronchitis; or sinonasal infection). A possible mold infection was defined as the presence of host factor and clinical criteria without mycological criteria.

Primary antifungal therapy was defined as the first therapy used upon diagnosis or suspicion of an IFI. A salvage therapy was considered any regimen administered after primary therapy. Clinical and radiologic findings were evaluated at baseline (diagnosis or suspicion of an IFI), week 4, week 6 and week 12 of follow-up. The galactomannan test for diagnosis purpose were considered if performed within 1 week of the start of therapy.

A breakthrough was defined as an infection occurring in a patient receiving prophylactically systemic antifungals with known activity against species causing his/her IFI for at least 7 consecutive days.

The response to therapy was evaluated at the end of primary therapy, at week 6 and at week 12 after primary therapy initiation. We classified response into complete response, partial response (improvement of clinical/radiologic findings but not completely resolved), failure (worsening of clinical/radiologic findings), relapse or stable. Death was evaluated during 6 months upon IFI diagnosis/suspicion.

Disparities were defined as having one of the following comorbidities criteria: age 65 years or more, obesity (BMI ≥ 30), diabetes mellitus, renal insufficiency (GFR < 50 mL/min/1.73 m), or prolonged QT interval (>450 ms in male and >470 in female).

### 2.3. Statistical Analysis

Categorical variables were compared using chi-square or Fisher’s exact test, as appropriate. Continuous variables were compared using Kruskal–Wallis test (for three-group comparisons) and Wilcoxon rank-sum test (for two-group comparisons). If a significant result (*p* < 0.05) was detected for a test that compared three groups, pairwise comparisons were performed with α levels adjusted using Holm’s sequential Bonferroni adjustment to control type I error. Cox proportional hazards regression model was used to identify the independent risk factors for mortality and evaluate the independent impact of the type of primary antifungal therapy on it. In addition, Kaplan–Meier method was used to evaluate the survival curves of patients with and without disparity and the log-rank test was used for a comparison. All tests were two-sided with a significance level of 0.05, except the pairwise comparisons with the α adjustment. Statistical analyses were performed using SAS version 9.4 (SAS Institute Inc, Cary, NC, USA).

## 3. Results

### 3.1. Demographics and Clinical Characteristics

We identified 112 oncologic malignancies and transplant patients, diagnosed with IFI. Three centers contributed to the cohort of this study: Brazil Center (51 patients), Spain Center (43 patients) and Switzerland Center (16 patients). The IFI diagnosis was definite in 29 patients, probable in 51 patients, possible in 23 patients and unknown in 9 patients. Patient epidemiological and clinical characteristics are presented in Table 1. The mean age was 55 years (14–77 years), and 63% (70 patients) were male. The most common underlying condition was lung transplant 44 (39%), followed by acute myeloid leukemia 37 (33%). Stem cell transplant was performed in 34 (31%) patients prior to or during IFI, and allogeneic transplantation was the main modality (28 patients). Neutropenia (ANC ≤ 500 cells/mL) was documented at the onset of IFI in 38% (42 patients), and 90% (37 patients) of them recovered from neutropenia during infection. A cumulative dose of steroids (prednisone equivalent) above 600 mg was received by 49 patients (69%). Seventeen patients were admitted to the ICU with IFI diagnosis.

### 3.2. Invasive Fungal Infections and Therapy

The etiological agents are represented in Table 1. Most cases were invasive aspergillosis 88 (79%). There were nine cases of *Fusarium* spp. (8%). In 15 patients (13%), another fungal infection was identified. The primary antifungal therapy was isavuconazole in 34 patients (30%), voriconazole in 35 patients (31%) and amphotericin B in 43 patients (38%). A total of 41 patients (37%) received combined therapy. The median duration of primary therapy was 53 days. Forty of them (36%) received salvage therapy. Table 2 shows each first-line therapeutic group according to epidemiological and clinical characteristics. The three groups have comparable characteristics concerning age, sex, SCT and GVHD incidence, and ICU admission at baseline. The Brazilian and Swiss centers had the majority of patients submitted to first-line amphotericin treatment (64% and 55%), while in the Spanish center, the main therapy was isavuconazole (65%). The patients with neutropenia at the onset of IFI were more likely to receive treatment with amphotericin than voriconazole and isavuconazole, respectively (Ampho vs. Isa: *p* < 0.0001; Ampho vs. Vori: *p* = 0.002). There was no difference regarding the choice between voriconazole or isavuconazole in the neutropenia setting. The three groups presented the same rate of recovery from neutropenia. Isavuconazole and voriconazole groups tended to receive the cumulative steroids dose above 600 mg (prednisone equivalent) during infection more often when compared to the Amphotericin group (*p* < 0.0001). The amphotericin group had a higher rate of definite or probable diagnosis of IFI when compared to the isavuconazole group [Isa (59%) vs. Ampho (91%): *p* = 0.001]. Patients with invasive pulmonary infection or sinus infection tended to receive amphotericin treatment more frequently when compared with isavuconazole and voriconazole (Ampho vs. Isa: *p* < 0.0001; Ampho vs. Vori: *p* = 0.023).

### 3.3. Disparities

Table 3 and Table 4 show the epidemiological and clinical characteristics of the population divided according to the presence or absence of one or more disparities criteria. The groups were homogeneous with respect to sex, underlying disease and incidence of SCT and GVHD. There was no difference between the incidence of neutropenia at the onset of IFI, recovery from neutropenia during infection and patients with a cumulative dose of steroids above 600 mg (prednisone equivalent) during infection. The ICU admission rate was the same in both groups. There was no difference between the treatment choice considering the presence of disparities. The adverse events related to primary therapy were the same.

### 3.4. Adverse Events

Twenty one percent of the patients presented adverse events related to primary therapy.

A total of 13% of the events resulted in drug modification (Table 1). The Isavuconazole group presented less adverse events when compared to voriconazole and amphotericin (Isa vs. Vori: *p* < 0.001; Isa vs. Ampho: *p* = 0.019-Table 2). The adverse events related to primary therapy drugs were the same in patients with or without disparities (Table 3), even when compared inside each treatment group (Table 4).

### 3.5. Outcomes

The mortality associated with IFI in 12 weeks was 12% (Table 1). In the three treatment groups, the favorable response to primary therapy was the same at 6 weeks, 12 weeks and end of therapy (Table 2). Similarly, there was also no difference in IFI-attributable death between any of the azole agents versus amphotericin at weeks 6 and weeks 12, respectively. On the other hand, by univariate analysis, the all-cause mortality was higher in the amphotericin group compared to the isavuconazole group at 6 weeks and 12 weeks, respectively (6 weeks: *p* = 0.008; 12 weeks: *p* = 0.002) and was higher in the amphotericin group compared to the voriconazole group at 12 weeks only (*p* = 0.016-Table 2). However, by Multivariable Cox regression analysis, this difference in all-cause mortality between the azoles and amphotericin could not be confirmed, as shown in Table 5. Fusarium infection and invasive pulmonary infection or sinus infection were the only independent risk factors associated with mortality. The clinical response to antifungal therapy, as well as all-cause and IFI-attributable deaths at weeks 6 and 12, were the same in both patients, with or without disparities (Table 3 and Table 4).

## 4. Discussion

IFI remains a large concern in high-risk patients, particularly following chemotherapy or transplant procedure. Over the last two decades, we have had a consistent improvement in the management of IFI with a well-standardized and reviewed consensus criteria of IFI, newly non-invasive diagnostic tools (biomarkers and computer tomography (CT) scan) and a new generation of azoles available [10,11,17,18,19,20,21,22,23]. However, the challenges in managing this infection in the setting of underlying malignancies and transplant populations continue. Liver toxicity, drug interaction, renal dysfunction, diabetes mellitus, the elderly and persistent neutropenia or lymphopenia are among the most common challenging scenarios in this population. The strength of our study is that it is an international multicenter protocol that included a large cohort of hematologic malignancy (HM) and transplant patients (SCT and lung transplant) with definite, probable or possible IFIs in which we compared the real-world clinical outcomes and safety of isavuconazole to voriconazole and amphotericin B. Another strength of our study is the disparity subset analysis.

Our cohort showed an elevated proportion of high-risk patients for IFI, with lung transplant, acute myeloid leukemia (AML) and allogeneic SCT recipients being the most common underlying disease. There was a significantly higher number of patients with AML treated with amphotericin B-based regimen compared to isavuconazole or voriconazole respectively. On the contrary, a higher proportion of lung transplant patients or patients on high-dose steroids prior to IFI were treated with isavuconazole and/or the voriconazole group. This distribution of the underlying disease, in the end, is quite homogeneous considering that those patients are under a high risk for IFI and mortality [1,2,24].

The most common fungal isolated were *Aspergillus* spp., followed by *Fusarium* spp., and the majority were definitive and probable IFI. The epidemiology of IFI over the last two decades is almost the same in existing studies [25,26,27]. However, in a manner different from our results, with high rates of invasive Aspergillosis, the current scenario in the era of new azoles anti-mold prophylaxis is a trend of a shift from aspergillus species to non-aspergillus species [17,28,29].

There was no difference in clinical response among the patients treated with any of the three compared anti-fungal agents. In our study, the favorable response to isavuconazole at 6 and 12 weeks, and at the end of treatment, was 90%. This is higher than the 45% response rate reported in the SECURE trial, and the 47% and 42% reported at 6 and 12 weeks, respectively, in previous studies [13,30].

Our real-world results reflect the findings of the SECURE study, whereby the isavuconazole showed non-inferior efficacy compared to voriconazole, and a lower all-cause mortality with zero deaths versus rates of 19% at 6 weeks and 29% at 12 weeks [13]. However, by multivariate analysis, only invasive fusarium infections and invasive pulmonary or sinus infections were associated with significantly higher all-cause mortality rates at 6 or 12 weeks. This is in line with other studies demonstrating high mortality associated with fusariosis [1,31].

Nowadays, one of the main concerns related to triazole use is the various toxicities such as the hepatic, dermatologic and neurotoxicities [32]. Our study showed that those patients who received isavuconazole were significantly less prone to develop related adverse events compared to those who received either voriconazole or amphotericin B-based regimen. Isavuconazole showed a very safe profile in terms of liver toxicity and hallucination compared to voriconazole, and lower rates of renal failure compared to amphotericin B-based regimen. Our results are consistent with what has been reported in the SECURE study [13].

Another concern related to triazole use are the drug-drug interactions with the targeted chemotherapies for hematological malignancies such as Ruxolitinib, Venetoclax and Gilteritinib, etc.; [33,34,35]. However, most new drugs undergo extensive hepatic metabolism and exhibit moderate to severe drug-drug interactions with triazole antifungal agents, commonly increasing the levels of targeted therapies in hematologic malignancy with severe side effects such as severe and prolonged neutropenia [36,37]. In the SCT setting, Letermovir, an antiviral CMV-specific drug, was recently approved for cytomegalovirus (CMV) prophylaxis in CMV IGG-positive patients [38]. Nevertheless, Letermovir can potentially affect the drug-metabolizing and clearance pathways of posaconazole and voriconazole, leading to decreased exposure [39,40]. To address these concerns, isavuconazole has demonstrated predictable and linear pharmacokinetics with low interpatient variability and a less complicated drug interaction profile, making it an attractive alternative in these circumstances [41,42,43]. Hence, the improved safety profile of Isavuconazole compared to voriconazole and amphotericin B in this current study is reassuring, particularly in this high-risk patient population.

Furthermore, treating patients with comorbidity, a typical scenario of high-risk patients for IFI, is always challenging to ensure that the drug delivered has good exposure and that the side effects are acceptable. Then, to explore this setting, we analyzed a composite of characteristics, called disparities, with the patient needing to have at least one of the five criteria (elderly patients, obesity, diabetes mellitus, patients with renal insufficiency and patients with prolonged QT interval) as described in the Section 2.

Our disparities sub-analysis shows that patients with one or more of the disparity criteria had a similar outcome (response to primary anti-fungal therapy, adverse events, all cause and IFI related mortality at 6 weeks and 12 weeks) to patients with no disparity criteria (Table 3 and Table 4). Thus, our data support the safety profile of these antifungal drugs particularly isavuconazole in the disparities population.

Facing the suitable characteristics of isavuconazole, its cost effectiveness compared with voriconazole is demonstrated in this current study and supported in the literature as an appropriate option for suspected invasive pulmonary aspergillosis [44,45,46,47].

The limitation of this study includes its retrospective nature. Due to the small number of patients with mucormycosis and fusariosis in the isavuconazole arm, the favorable invasive aspergillosis results should be extrapolated with caution to the other pathogenic fungi.

## 5. Conclusions

In the real-world treatment experience of IFI in patients with underlying malignancy and transplant patients, isavuconazole was associated with the best safety profile compared to voriconazole or amphotericin B-based regimen. However, by multivariate analysis, isavuconazole, voriconazole or amphotericin B had a comparable outcome in this high-risk patient population. However, invasive fusariosis and invasive pulmonary or sinus fungal infections were the only factors independently associated with poor outcomes (higher rates of all-cause mortality at 6 or 12 weeks). Disparity criteria did not affect the response to anti-fungal therapy and overall outcome, including mortality.

## Figures and Tables

**Table 1 jof-09-00166-t001:** Patient characteristics, treatment and outcomes.

Variables	Patients
	(*n* = 112)
	*N* (%)
Center	
Brazil	51 (46)
Spain	43 (38)
Switzerland	18 (16)
Age (years), median (range)	55 (14–77)
Sex, male	70 (63)
Underlying diseases	
Lung transplant	44 (39)
AML	37 (33)
ALL	6 (5)
CLL	4 (4)
CML	3 (3)
Lymphoma	10 (9)
Myeloma	2 (2)
Solid tumor	6 (5)
SCT prior to or during IFI	34/111 (31)
Type of SCT	
Autologous	6/34 (18)
Allogeneic	28/34 (82)
Neutropenia at the onset of IFI	42/111 (38)
Recovery from neutropenia during infection	37/41 (90)
Steroid treatment	72/111 (65)
Cumulative dose of steroids	
≥600 mg (prednisone equivalent)	49/71 (69)
<600 mg (prednisone equivalent)	22/71 (31)
ICU at baseline	17/109 (16)
Diagnosis of IFI	
Definite	29 (26)
Probable	51 (46)
Possible	32 (28)
Organism of IFI	
*Aspergillus*	88 (79)
*Fusarium*	9 (8)
*Mucor*	7 (6)
*Trichosporon* spp.	3 (3)
Others	5 (4)
Positive fungal culture	53/98 (54)
*Aspergillus* spp.	35 */52 (67)
*Fusarium* spp.	5/52 (10)
*Mucor* spp.	2 ^#^/52 (4)
*Candida* spp.	2/52 (4)
*Trichosporon* spp.	2/52 (4)
*Rhizopus* spp.	2 ^#^/52 (2)
*Scedosporium apiospermum*	2/52 (4)
*Lichteimia*	1/52 (2)
*Cryptococcus laurentii*	1/52 (2)
*Magnusiomyces capitatus*	1/52 (2)
Primary therapy	
Isavuconazole	34 (30)
Voriconazole	35 (31)
Amphotericin B	43 (38)
Study drug used in primary therapy	
Alone	71 (63)
In combination	41 (37)
Duration of primary therapy (days), median (IQR)	53 (21–140)
Receiving salvage therapy ^&^	40 (36)
Amphotericin B	11/40 (28)
Itraconazole	1/40 (3)
Voriconazole	21/40 (53)
Posaconazole	1/40 (3)
Isavuconazole	12/40 (30)
Echinocandins	4/40 (10)
Response to primary therapy	
At week 6	
Complete response	38 (34)
Partial response	41 (37)
Failure	7 (6)
Stable	7 (6)
Unknown or non-applicable (primary therapy ended before week 6)	19 (17)
At week 12	
Complete response	46 (41)
Partial response	26 (23)
Failure	5 (4)
Stable	4 (4)
Relapse	1 (1)
Unknown or non-applicable (primary therapy ended before week 12)	30 (27)
At end of therapy	
Complete response	65 (58)
Partial response	12 (11)
Failure	16 (14)
Stable	4 (4)
Relapse	1 (1)
Unknown	14 (13)
Adverse events related to primary therapy	23 (21)
≥2 LAK phosphatase	6 (5)
≥2 Bilirubin	3 (3)
≥Creatinine	3 (3)
≥SGPT	9 (8)
High LFTS	1 (1)
Renal failure	5 (4)
Hallucination	3 (3)
x 2 GAMMA GT	6 (5)
Altered level of consciousness	1 (1)
Adverse events resulting in drug modification	15 (13)
Adverse events resolved after drug modification	12/14 (86)
Mortality since IFI diagnosis	
All-cause death at week 6	10/110 (9)
IFI-attributable death at week 6	8/110 (7)
All-cause death at week 12	17/110 (15)
IFI-attributable death at week 12	13/110 (12)

Note: * One patient infected with *Aspergillus* spp. was also infected with *Pascylomyces*. ^#^ One patient was infected by both of *Mucor* and *Rhizopus* spp. ^&^ Eight patients received more than one antifungal agents as salvage therapy. For any variable with data missing, the number of patients with data available for this variable was added as the denominator.

**Table 2 jof-09-00166-t002:** Primary therapy comparison.

Characteristics and Outcomes	Isavuconazole (G1)	Voriconazole (G2)	Amphotericen B (G3)	*p*-Value	Pairwise Comparisons with Significant Differences ^#^
	(*n* = 34)	(*n* = 35)	(*n* = 43)		
	*N* (%)	*N* (%)	*N* (%)		
Age (years), median (range)	60 (20–76)	44 (18–72)	54 (14–77)	0.09	
Sex, male	22 (65)	19 (54)	29 (67)	0.47	
Underlying disease				0.001	G1 vs. G3: *p* < 0.001; G2 vs. G3: *p* = 0.013
AML	5 (15)	9 (26)	23 (53)		
Others	29 (85)	26 (74)	20 (47)		
BMT—Allogeneic	5 (15)	10/34 (29)	13 (30)	0.24	
GVHD	2/33 (6)	5/33 (15)	5/37 (14)	0.54	
Neutropenia at the onset of IFI	5 (15)	10 (29)	27/42 (64)	<0.0001	G1 vs. G3: *p* < 0.0001; G2 vs. G3: *p* = 0.002
Recovery from neutropenia during infection	4/5 (80)	10/10 (100)	23/26 (88)	0.29	
Cumulative steroids ≥600 mg (prednisone equivalent) during infection	26 (76)	18/33 (55)	5 (12)	<0.001	G1 vs. G3: *p* < 0.0001; G2 vs. G3: *p* < 0.0001
ICU at baseline	3/33 (9)	6/34 (18)	8/42 (19)	0.46	
Diagnosis of IFI				0.015	G1 vs. G3: *p* = 0.005;
Definite	8/29 (28)	6/31 (19)	15 (35)		
Probable	9/29 (31)	18/31 (58)	24 (56)		
Possible	12/29 (41)	7/31 (23)	4 (9)		
Organism of IFI					
*Aspergillus*	30 (88)	31 (89)	27 (63)	0.003	G1 vs. G3: *p* = 0.012; G2 vs. G3: *p* = 0.01
*Fusarium*	0 (0)	0 (0)	9 (21)	0.0001	G1 vs. G3: *p* = 0.004; G2 vs. G4: *p* = 0.004
*Mucor*	1 (3)	0 (0)	6 (14)	0.023	None
*Trichosporon* spp.	2 (6)	1 (3)	0 (0)	0.20	
Others	1 (3)	3 (9)	1 (2)	0.52	
Invasive pulmonary infection or sinus infection	12/33 (36)	20 (57)	34/42 (81)	<0.001	G1 vs. G3: *p* < 0.0001; G2 vs. G3: *p* = 0.023
Favorable response to primary therapy					
At week 6	27/30 (90)	23/29 (79)	29/34 (85)	0.51	
At week 12	26/29 (90)	21/26 (81)	25/27 (93)	0.40	
At end of primary therapy	27/30 (90)	23/30 (77)	27/38 (71)	0.16	
Adverse events related to primary therapy drug	1 (3)	12 (34)	10 (23)	0.005	G1 vs. G2: *p* < 0.001; G1 vs. G3: *p* = 0.019
≥2 LAK phosphatase	0 (0)	6 (17)	0 (0)		
≥2 Bilirubin	0 (0)	3 (9)	0 (0)		
≥Creatinine	0 (0)	0 (0)	3 (7)		
≥SGPT	0 (0)	8 (23)	1 (2)		
High LFTS	0 (0)	1 (3)	0 (0)		
Renal failure	0 (0)	0 (0)	5 (12)		
Hallucination	1 (3)	1 (3)	1 (2)		
x 2 GAMMA GT	0 (0)	6 (17)	0 (0)		
Altered level of consciousness	0 (0)	1 (14)	0 (0)		
Mortality since IFI diagnosis					
All-cause death at week 6	0/33 (0)	2 (6)	8/42 (19)	0.011	G1 vs. G3: *p* = 0.008
IFI-attributable death at week 6	0/33 (0)	2 (6)	6/42 (14)	0.06	
All-cause death at week 12	1/33 (3)	3 (9)	13/42 (31)	0.002	G1 vs. G3: *p* = 0.002; G2 vs. G3: *p* = 0.016
IFI-attributable death at week 12	1/33 (3)	2 (6)	10/42 (24)	0.014	None

Note: ^#^ When a *p*-value < 0.05 was found from a global test comparing the three groups, pairwise comparisons were performed to locate all the significant differences. The α levels were adjusted using Holm’s sequential Bonferroni adjustment to control type I error. For any variable with data missing or data non-applicable, the number of patients with data available for this variable was added as the denominator.

**Table 3 jof-09-00166-t003:** Comparing patients with and without disparity.

Characteristics and Outcomes	Non-Disparity	Disparity *	*p*-Value
	(*n* = 46)	(*n* = 66)	
	*N* (%)	*N* (%)	
Age (years), median (range)	46 (14–67)	60 (17–77)	<0.0001
Sex, male	27 (59)	43 (65)	0.49
Underlying disease			0.09
AML	11 (24)	26 (39)	
Others	35 (76)	40 (61)	
SCT—Allogeneic	9 (20)	19/65 (29)	0.25
GVHD	7/44 (16)	5/59 (8)	0.24
Neutropenia at the onset of IFI	13/45 (29)	29 (44)	0.11
Recovery from neutropenia during infection	12/13 (92)	25/28 (89)	>0.99
Cumulative steroids ≥ 600 mg (prednisone equivalent) during infection	22/45 (49)	27/65 (42)	0.45
ICU at baseline	8 (17)	9/63 (14)	0.66
Diagnosis of IFI			
Definite	12/41 (29)	17/62 (27)	
Probable	22/41 (54)	29/62 (47)	
Possible	7/41 (17)	16/62 (26)	
Organism of IFI			
*Aspergillus*	36 (78)	52 (79)	0.95
*Fusarium*	2 (4)	7 (11)	0.30
*Mucor*	2 (4)	5 (8)	0.70
*Trichosporon* spp.	2 (4)	1 (2)	0.57
Others	4 (9)	1 (2)	
Invasive pulmonary infection or sinus infection	27 (59)	39/64 (61)	0.81
Primary therapy			0.32
Isavuconazole	12 (26)	22 (33)	
Voriconazole	18 (39)	17 (26)	
Amphotericin B	16 (35)	27 (41)	
Favorable response to primary therapy			
At week 6	31/38 (82)	48/55 (87)	0.45
At week 12	28/32 (88)	44/50 (88)	>0.99
At end of primary therapy	30/39 (77)	47/59 (80)	0.75
Adverse events related to primary therapy drug	10 (22)	13 (20)	0.79
Mortality since IFI diagnosis			
All-cause death at week 6	3/45 (7)	7/65 (11)	0.52
IFI-attributable death at week 6	3/45 (7)	5/65 (8)	>0.99
All-cause death at week 12	6/45 (13)	11/65 (17)	0.61
IFI-attributable death at week 12	5/45 (11)	8/65 (12)	0.85

Note * A patient with disparity had to have at least one of the following features: age ≥ 65 years, obesity (BMI ≥ 30), diabetes mellitus, renal insufficiency or prolonged QT interval. For any variable with data missing or data non-applicable, the number of patients with data available for this variable was added as the denominator.

**Table 4 jof-09-00166-t004:** Comparing patients with and without disparity by primary therapies.

Characteristics and Outcomes	Isavuconazole			Voriconazole			Amphotericin B		
	Non-Disparity	Disparity	*p*-Value	Non-Disparity	Disparity	*p*-Value	Non-Disparity	Disparity	*p*-Value
	(*n* = 12)	(*n* = 22)		(*n* = 18)	(*n* = 17)		(*n* = 16)	(*n* = 27)	
	*N* (%)	*N* (%)		*N* (%)	*N* (%)		*N* (%)	*N* (%)	
Age (years), median (range)	56 (31–67)	63 (20–76)	0.15	40 (18–62)	58 (27–72)	0.03	44 (14–63)	60 (17–77)	0.008
Sex, male	7 (58)	15 (68)	0.71	11 (61)	8 (47)	0.40	9 (56)	20 (74)	0.23
Underlying disease			0.14			>0.99			
AML	0 (0)	5 (23)		5 (28)	4 (24)		6 (38)	17 (63)	0.11
Others	12 (100)	17 (77)		13 (72)	13 (76)		11 (69)	16 (59)	
SCT—Allogeneic	0 (0)	5 (23)	0.14	6 (33)	4/16 (25)	0.71	3 (19)	10 (37)	0.31
GVHD	0 (0)	2/21 (10)	0.52	5 (28)	0/15 (0)	0.049	2/14 (14)	3/23 (13)	>0.99
Neutropenia at the onset of IFI	0 (0)	5 (23)	0.14	5 (28)	5 (29)	>0.99	8/15 (53)	19 (70)	0.27
Recovery from neutropenia during infection	NA	4/5 (80)		5/5 (100)	5/5 (100)		7/8 (88)	16/18 (89)	>0.99
Cumulative steroids ≥ 600 mg (prednisone equivalent) during infection	12 (100)	14 (64)	0.03	10/17 (59)	8/16 (50)	0.61	0 (0)	5 (19)	0.14
ICU at baseline	2 (17)	1/21 (5)	0.54	3 (17)	3/16 (19)	>0.99	3 (19)	5/26 (19)	>0.99
Diagnosis of IFI			0.78			0.89			0.41
Definite	3/9 (33)	5/20 (25)		3/16 (19)	3/15 (20)		6 (38)	9 (33)	
Probable	2/9 (22)	7/20 (35)		10/16 (63)	8/15 (53)		10 (63)	14 (52)	
Possible	4/9 (44)	8/20 (40)		3/16 (19)	4/15 (27)		0 (0)	4 (15)	
Organism of IFI									
*Aspergillus*	10 (83)	20 (91)	0.60	14 (78)	17 (100)	0.10	12 (75)	15 (56)	0.2
*Fusarium*	0	0		0	0		2 (13)	7 (26)	0.45
*Mucor*	1 (8)	0 (0)	0.35	0	0		1 (6)	5 (19)	0.39
*Trichosporon* spp.	1 (8)	1 (5)	>0.99	1 (6)	0 (0)	>0.99	0	0	
Others	0 (0)	1 (5)	>0.99	3 (17)	0 (0)	0.23	1 (6)	0 (0)	0.37
Invasive pulmonary infection or sinus infection	2 (17)	10/21 (48)	0.13	11 (61)	9 (53)	0.63	14 (88)	20/26 (77)	0.69
Favorable response to primary therapy									
At week 6	10/10 (100)	17/20 (85)	0.53	10/14 (71)	13/15 (87)	0.39	11/14 (79)	18/20 (90)	0.63
At week 12	9/9 (100)	17/20 (85)	0.53	10/13 (77)	11/13 (85)	>0.99	9/10 (90)	16/17 (94)	>0.99
At end of primary therapy	10/10 (100)	17/20 (85)	0.53	8/14 (57)	15/16 (94)	0.031	12/15 (80)	15/23 (65)	0.47
Adverse events related to primary therapy drug	0 (0)	1 (5)	>0.99	7 (39)	5 (29)	0.55	3 (19)	7 (26)	0.72
Mortality since IFI diagnosis									
All-cause death at week 6	0 (0)	0/21 (0)		2 (11)	0 (0)	0.49	1/15 (7)	7 (26)	0.22
IFI-attributable death at week 6	0 (0)	0/21 (0)		2 (11)	0 (0)	0.49	1/15 (7)	5 (19)	0.40
All-cause death at week 12	0 (0)	1/21 (5)	>0.99	3 (17)	0 (0)	0.23	3/15 (20)	10 (37)	0.31
IFI-attributable death at week 12	0 (0)	1/21 (5)	>0.99	2 (11)	0 (0)	0.49	3/15 (20)	7 (26)	>0.99

Abbreviation: NA = Non-Applicable. Note: A patient with disparity had to have at least one of the following features: age ≥ 65 years, obesity (BMI ≥ 30), diabetes mellitus, renal insufficiency or prolonged QT interval. For any variable with data missing or data non-applicable, the number of patients with data available for this variable was added as the denominator.

**Table 5 jof-09-00166-t005:** Multivariable Cox regression analysis of risk factors for mortality.

Independent Risk Factor	Adjusted HR	95% CI	*p*-Value
6-week mortality			
Organism of IFI			0.003
Fusarium	8.03	2.09 to 30.94	
Other fungal species	Reference		
Type of IFI infection			0.042
Invasive pulmonary infection or sinus infection	6.70	1.07 to 42.00	
Other infection	Reference		
12-Week mortality			
Organism of IFI			
Fusarium	8.15	2.64 to 25.18	<0.001
Other fungal species	Reference		
Type of IFI infection			
Invasive pulmonary infection or sinus infection	12.81	2.23 to 73.62	0.004
Other infection	Reference		

Abbreviations: Adjusted HR = Adjusted Hazard Ratio; 95% CI = 95% Confidence Interval. Independent impact of primary therapy on 6- and 12-week mortality. After adjusting for the independent risk factors, we identified for 6- and 12-week mortality, with the type of primary fungal therapy (isavuconzole vs. voriconazole vs. Amphotericin B) showing no significant impact on 6-week (*p* = 0.51) and 12-week mortality.

## Data Availability

Not applicable.

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
