# Peer review of "Comparing the Real-World Use of Isavuconazole to Other Anti-Fungal Therapy for Invasive Fungal Infections in Patients with and without Underlying Disparities: A Multi-Center Retrospective Study"

_jof, 2023, doi:10.3390/jof9020166_

Round 1

Reviewer 1 Report

In the paper entitled "Comparing the Real-World Use of Isavuconazole to Other Anti-Fungal Therapy for Invasive Fungal Infections in Patients with and without Underlying Disparities: A Multi-Center Retrospective Study", the authors present a retrospective tricentered study regarding the use of isavuconazole as primary therapy of Invasive Fungal Infections, mostly in immuno-compromised patients, in comparison with the use of voriconazole and amphotericin B.

Considering the rising incidence of IFIs, affecting not only immuno-compromised patients, the idea of this study is a great one. Including patients from more than one center/country/continent is also a good idea, especially when there is more then one primary therapeutic option.

The study and paper are both well organized, however, the paper still needs some minor to major corrections before acceptance in JoF:

- Major English language editing (the paper's major drawback)

-the text must be Justified to the right

- page 1 - line 46: replace "fusarium" with "Fusarium"

- all fungal strains, expressed as Genus species must be in Italic face (eg. Aspergillus spp.)

- there are many spaces missing or in plus (depends on the phrase)

- page 2 - line 61: I suggest the authors to delete "the effect" since for the 3 antifungals discussed the antifungal effect is great, only the use is limited because of sometimes poor safety profile

- page 2 - line 67: replace "drug interactions with other drugs" with "interactions with other drugs" or "drug interactions"

- page 3 - line 97: replace "isavuconazole, voriconazole and amphotericin" with  "isavuconazole, voriconazole or amphotericin" , since only one of the 3 antifungals was administered

- page 4 - line 166" replace "Lung" with "lung"

- page 4 - paragraph 3.2. : I was wondering if the authors can somehow justify the choice of amphotericin (considering its renal toxicity) as first line therapy in Brazil and Switzerland?

- all abbreviations should be defined when first used in the text.

Author Response

Response to Reviewer 1 Comments

Point 1: Major English language editing (the paper's major drawback)

Response 1: The English language was reviewed and adjusted. Please, if you have a specific sentence or paragraph that you want to be editing, please, let us know.

Point 2: The text must be Justified to the right

Response 2: The text was Justified to the right as suggested.

Point 3: page 1 - line 46: replace "fusarium" with "Fusarium"

Response 3: The word was adjusted as suggested, page 1 - line 46.

Point 4: all fungal strains, expressed as Genus species must be in Italic face (eg. Aspergillus spp.)

Response 4: The fungal strains, expressed as Genus species, were adjusted in the Italic face as suggested.

Point 5: There are many spaces missing or in plus (depends on the phrase)

Response 5: The spaces missing or in plus were adjusted as suggested.

Point 6: page 2 - line 61: I suggest the authors to delete "the effect" since for the 3 antifungals discussed the antifungal effect is great, only the use is limited because of sometimes poor safety profile

Response 6: We agree with the comment and addressed as suggested, page 2 - line 62.

Point 7: page 3 - line 97: replace "isavuconazole, voriconazole and amphotericin" with “isavuconazole, voriconazole or amphotericin”, since only one of the 3 antifungals was administered

Response 7: We agree with the comment, and addressed as suggested, page 3 - line 106.

Point 8: page 4 - line 166" replace "Lung" with "lung"

Response 8: We addressed as suggested, page 4 - line 209.

Point 9: page 4 - paragraph 3.2. : I was wondering if the authors can somehow justify the choice of amphotericin (considering its renal toxicity) as first line therapy in Brazil and Switzerland?

Response 9: Both centers use only liposomal amphotericin B, in regular dose (3mg/kg), that has less renal toxicity compared to amphotericin B Deoxycholate. We added the information in the inclusion criteria section, page 3, line 106.

Point 10: all abbreviations should be defined when first used in the text.

Response 10: All abbreviations were adjusted as suggested

Reviewer 2 Report

The authors of the manuscript titled "Comparing the real-world use of isavuconazole to other anti-fungal therapy for invasive fungal infections in patients with and without underlying disparities: A multi-center retrospective study" report the better safety profile of isavuconazole compared to voriconazole and amphotericin B for patients with invasive fungal infections. The body of work presented here is appropriate for the Journal of Fungi; however, it needs minor revisions before it can be considered for publication.

Points that need to be addressed.

  1.  The authors should fix the abstract. Please remove the title's background, methods, results, and conclusions. The abstract need not be such detailed, and makes sure it is compact and to the point.
  2.  On page 2, line 71, it should read approved.
  3. In the conclusions section, line 384, please remove inf after invasive fusariosis.

Author Response

Response to Reviewer 2 Comments

Point 1: The authors should fix the abstract. Please remove the title's background, methods, results, and conclusions. The abstract need not be such detailed and makes sure it is compact and to the point.

Response 1:  The abstract was adjusted as suggested.

Point 2: On page 2, line 71, it should read approved.

Response 2: The word was adjusted to approved on page 2, line 75.

Point 3: In the conclusions section, line 384, please remove inf after invasive fusariosis.

Response 3: The ïnf” was deleted on page 20, line 452.

Reviewer 3 Report

The manuscript is a retrospective study comparing the safety and efficacy of the new antifungal drug Isavuconazule which belongs to the azole class of antifungal to another drug in the azole group (Voriconazole) and a drug in a different class (the polyene group) Amphotericin B, considered by some as the model antifungal drug but limited in clinical use by its side effects and lack or oral preparation.

The authors define their terms and use statistical analysis to combine data from 3 medical centres from different countres to show the non-inferiorty of the efficacy of Isacanazole compared Voriconazole which is recommeded for first line treatment of invasive fungal infection and Amphoteric B. The authors also show that has a better safety profile compared to the other two.

A very good study addresssing a pertinent clinical question.

There are a few points authors must address

1. The use of the word disparity- health disparity is a specific term and the use in this manuscript does not seem appopriate. The authors correctly used the term comorbidities before definining the term and thereafter used disparity for patient that had comorbidities . I will suggest the authors rathe use the term comorbidities throughout the manuscrpt.

2, In the abstrct the authors mentioned that the patients were aged 40-63 years, however, under the results and in table 1 the age was 14-77 year. Could the authors please explain this disparith

3. There is the need for some spelling editting or english correction e,g, in line 71 the word approved was spelt as spproved again in line 76 the word both appears inappropriate as they authrs describe three and not two organization guidelines (ESCMID, ECIL-6 and IDA) furthermore, under line 203 the phrase with respect to would be more appropirate than concerning. Could the authos please let a native engish speaker check the script for things like this and point uner 1, above.

4. The are some disparities between number used and those in tables e,g, table 1 uder patients with SCT prior to or during IFI they present 34/111 but our study population is 112, the same situation is present for nutropenia at onset of IFI and cumulative dose of IFI. Was all this data mising for 1 patient then why did we include?. For ICU admussion at baseline only 109 patient were accounted for what happened to the last 3?

5. For our study population the authors include 9 patient with unknown or undetermined diagnosis of IFI- why were these patients included in the study?

6. Under the introduction the manuscript mentions that the best antifungal to be used may be influenced by ethnicity or comobidities as ethnicity may be easy to obtain in a retrospective study why was this not studies?

7. Could the authors please explain which patient are those with pulmonary or sinus infections are thes patient without proven infections but probable and possible IFI with disease localized to these site by imaging?

Thank you

Author Response

Response to Reviewer 3 Comments

Point 1: The use of the word disparity- health disparity is a specific term and the use in this manuscript does not seem appopriate. The authors correctly used the term comorbidities before definining the term and thereafter used disparity for patient that had comorbidities . I will suggest the authors rathe use the term comorbidities throughout the manuscrpt.

Response 1:  We agree and adjusted defining “Disparities” as having one of the following comorbidities criteria: age 65 years or more, obesity (BMI>=30), diabetes mellitus, renal insufficiency (GFR<50mL/min/1.73m), or prolonged QT interval (>450ms in male and >470 in female). On page 4, line 183

Point 2: In the abstract the authors mentioned that the patients were aged 40-63 years, however, under the results and in table 1 the age was 14-77 year. Could the authors please explain this disparith

Response 2: There was a mistake in the abstract. We adjusted to 14-77 years on page 1, line 35.

Point 3: There is the need for some spelling editting or english correction e,g, in line 71 the word approved was spelt as spproved again in line 76 the word both appears inappropriate as they authrs describe three and not two organization guidelines (ESCMID, ECIL-6 and IDA) furthermore, under line 203 the phrase with respect to would be more appropirate than concerning. Could the authos please let a native engish speaker check the script for things like this and point uner 1, above.

Response 3: we addressed all changes as suggested.

  • ”In line 71 the word approved was spelt as spproved” the word was adjusted; line 72;
  • “line 76 the word both appears inappropriate” the word both was deleted and added “the”, line 77;
  • “line 203 the phrase with respect to would be more appropirate than concerning, we replace as suggested, line 249.

Additionaly, a native speaker, reviewed all manuscript.

Point 4: The are some disparities between number used and those in tables e,g, table 1 uder patients with SCT prior to or during IFI they present 34/111 but our study population is 112, the same situation is present for nutropenia at onset of IFI and cumulative dose of IFI. Was all this data mising for 1 patient then why did we include?. For ICU admussion at baseline only 109 patient were accounted for what happened to the last 3?

Response 4:  The missing data for different variables came from different patients. For instance, in Table 1, the patient with data missing for “SCT prior to or during IFI” (n=1) had data about “Neutropenia at onset of IFI”, and the patient with missing data for “Neutropenia at onset of IFI” (n=1) had data about “SCT prior to or during IFI”. They were two different patients. Given that patients who did not have 100% complete data still could contribute a lot of useful data to this study and the small sample size of each treatment group in our study, we decided to keep all the patients to maintain our statistical power as much as possible. Regarding ICU status, unfortunately, 3 patients’data are unknown. This is understandable due to the nature of this study being retrospective, which had been listed as one study limitation in the manuscript.

For any variables with data missing, we added denominators in Tables for a clarification on how many patients were evaluable for this specific variable. For instance, “34/111 for SCT prior to or during IFI” means totally 111 patients were evaluable for this variable and of them 34 had SCT. Following the reviewer’s comment, we have added a footnote under all the applicable tables to explain such data expression.

Point 5: For our study population the authors include 9 patient with unknown or undetermined diagnosis of IFI- why were these patients included in the study?

Response 5: There was a mistake, all of these cases were possible IFI. The numbers were adjusted in table 1.

Point 6: Under the introduction the manuscript mentions that the best antifungal to be used may be influenced by ethnicity or comobidities as ethnicity may be easy to obtain in a retrospective study why was this not studies?

Response 6: We deleted ethnicity on page 2, line 32, because there is no or a weak correlation between the current antifungal study drugs and ethnicity.

Point 7: Could the authors please explain which patient are those with pulmonary or sinus infections are thes patient without proven infections but probable and possible IFI with disease localized to these site by imaging?

Response 7: Of the 66 patients with invasive pulmonary infection or sinus infections, 52 had probable or possible IFIs.

Reviewer 4 Report

This interesting study its aims to compare the real-world clinical outcomes and safety of Isavuconazole to Voriconazole and Amphotericin B based regimen in patients with underlying malignancies and transplant. The study is well structured; however, there are certain details that I consider can further enrich the presentation.

Minor corrections

1. Write scientific names in italics

2. Change Lichteinia to Lichteimia

3. Please, indicate the inclusion and exclusion criteria for the study

4. It would be convenient if they indicated the doses of treatments, as well as the species of the causative agents of the IFIs

5. References should be written according to the instructions for authors (JoF)

Author Response

Response to Reviewer 4 Comments

Point 1: Write scientific names in italics

Response 1: All scientific names are now in italics

Point 2: Change Lichteinia to Lichteimia

Response 2: The text was Justified to the right as suggested.

Point 3: Please, indicate the inclusion and exclusion criteria for the study

Response 3: The inclusion and exclusion criteria were added.

Point 4: It would be convenient if they indicated the doses of treatments, as well as the species of the causative agents of the IFIs

Response 4: The information about the doses of treatments was added on page 3, lines 101-103, and for the species of the causative agents, we have only the Genus species that are listed on Table 1.

Point 5: References should be written according to the instructions for authors (JoF)

Response 5: The references were adjusted as suggested.